# Impact of Different Cell Counting Methods in Molecular Monitoring of Chronic Myeloid Leukemia Patients

**DOI:** 10.3390/diagnostics12051051

**Published:** 2022-04-22

**Authors:** Stefania Stella, Silvia Rita Vitale, Fabio Stagno, Michele Massimino, Adriana Puma, Cristina Tomarchio, Maria Stella Pennisi, Elena Tirrò, Chiara Romano, Francesco Di Raimondo, Emma Cacciola, Rossella Cacciola, Livia Manzella

**Affiliations:** 1Department of Clinical and Experimental Medicine, University of Catania, 95123 Catania, Italy; silviarita.vitale@gmail.com (S.R.V.); michedot@yahoo.it (M.M.); adry.p88@hotmail.it (A.P.); cristina.tomarchio@hotmail.it (C.T.); perny76@gmail.com (M.S.P.); chiararomano83@gmail.com (C.R.); rcacciol@unict.it (R.C.); manzella@unict.it (L.M.); 2Center of Experimental Oncology and Hematology, A.O.U. Policlinico “G.Rodolico-San Marco”, 95123 Catania, Italy; ele_tir@yahoo.it; 3Division of Hematology and Bone Marrow Transplant, A.O.U. Policlinico “G.Rodolico-San Marco”, 95123 Catania, Italy; fsematol@tiscali.it (F.S.); diraimon@unict.it (F.D.R.); 4Department of Surgical, Oncological and Stomatological Sciences, University of Palermo, 90127 Palermo, Italy; 5Department of Surgery, Medical and Surgical Specialities, Section of Hematology, University of Catania, 95123 Catania, Italy; 6Department of Medical, Surgical Sciences and Advanced Technologies “G.F. Ingrassia”, University of Catania, 95123 Catania, Italy; ecacciol@unict.it; 7Hemostasis/Hematology Unit, A.O.U. Policlinico “G.Rodolico-San Marco”, 95123 Catania, Italy

**Keywords:** chronic myeloid leukemia, cell count, Q-PCR, *BCR-ABL1/ABL1*

## Abstract

Background: Detection of *BCR-ABL1* transcript level via real-time quantitative-polymerase-chain reaction (Q-PCR) is a clinical routine for disease monitoring, assessing Tyrosine Kinase Inhibitor therapy efficacy and predicting long-term response in chronic myeloid leukemia (CML) patients. For valid Q-PCR results, each stage of the laboratory procedures need be optimized, including the cell-counting method that represents a critical step in obtaining g an appropriate amount of RNA and reliable Q-PCR results. Traditionally, manual or automated methods are used for the detection and enumeration of white blood cells (WBCs). Here, we compared the performance of the manual counting measurement to the flow cytometry (FC)-based automatic counting assay employing CytoFLEX platform. Methods: We tested five different types of measurements: one manual hemocytometer-based count and four FC-based automatic cell-counting methods, including absolute, based on beads, based on 7-amino actinomycin D, combining and associating beads and 7AAD. The recovery efficiency for each counting method was established considering the quality and quantity of total RNA isolated and the Q-PCR results in matched samples from 90 adults with CML. Results: Our analyses showed no consistent bias between the different types of measurements, with comparable number of WBCs counted for each type of measurement. Similarly, we observed a 100% concordance in the amount of RNA extracted and in the Q-PCR cycle threshold values for both *BCR-ABL1* and *ABL1* gene transcripts in matched counted specimens from all the investigated groups. Overall, we show that FC-based automatic absolute cell counting has comparable performance to manual measurements and allows accurate cell counts without the use of expensive beads or the addition of the time-consuming intercalator 7AAD. Conclusions: This automatic method can replace the more laborious manual workflow, especially when high-throughput isolations from blood of CML patients are needed.

## 1. Introduction

Chronic Myeloid Leukemia (CML) is a hematological disorder characterized by the neoplastic transformation of a hematopoietic stem cell carrying the Philadelphia (Ph) chromosome that juxtaposes the *breakpoint cluster region* (*BCR)* and the *Abelson1* (*ABL1)* genes [1,2,3]. The ensuing Ph chromosome, at the molecular level, leads to the formation of the *BCR-ABL1* fusion oncogene encoding for multi-domain *BCR-ABL1* oncoproteins with constitutive tyrosine kinase activity that induces aberrant activation of several intracellular pathways driving malignant transformation [4,5,6]. 

Over the past 20 years, the development of *BCR-ABL1* tyrosine kinase inhibitors (TKIs) has significantly improved the outcomes of most CML patients, generating unprecedent rates of complete hematological (CHR), cytogenetic (CCyR) and molecular (MR) responses [7,8,9,10]. Despite these excellent results, around 30–50% of CML patients failed to achieve an optimal response (OR) as defined by the current European Leukemia Net (ELN) recommendations [11,12]. In this context, the identification of a resistance mechanism becomes critical to defining the management of this group of individuals, eligible to switch to another TKI [13,14,15,16]. At the same time, patients with persistent deep molecular response, after TKIs discontinuation, may be considered for treatment free-remission (TFR) [17,18,19]. On the bases of this observation a workflow generating high quality molecular data is mandatory for therapeutic decision-making. 

In the current clinical practice, the quantitative polymerase chain reaction (Q-PCR) is the “gold standard” for diagnostic *BCR-ABL1* transcript monitoring [20,21,22,23]. Overall, the evaluation of *BCR-ABL1* oncogene transcript should be performed every 3 months after TKI therapy initiation, then at least every 3–6 months. For valid Q-PCR data, standardization of each stage of the laboratory procedures is mandatory, including the amount of blood collected, the method for measuring total white blood cells (WBCs), and RNA isolations [24,25,26,27]. Particularly, the quantity of isolated WBCs and total RNA extracted are indispensable requisites for the accuracy and reproducibility of Q-PCR analyses. To this purpose, cell-counting measurement represents a critical step. Despite its common use, counting of cells with high accuracy and precision often remains an issue. An error in cell counting will propagate through subsequent procedures and will affect the overall quality and quantity of *BCR-ABL1* and *ABL1* gene transcripts. 

The considerable burden of work for molecular laboratories necessitates the development of a high-throughput method for a fast, reproducible, and efficient counting assay. Traditionally, manual counting-chambers or automated methods are used for the detection and enumeration of cells. Although the traditional way of counting cells manually, by hemocytometer, is very simple and straightforward, it is often error-prone, time-consuming, and highly subjective. For example, a wrong dilution or wrong samples may result in inappropriate volumes and concentrations of cells. Therefore, an ideal cell counting method should be accurate as well as repeatable, with low variability attributed to operator error [28]. Due to the ability to analyze single cells or particles of different sizes, flow cytometry is particularly suited to cell counting. Common methods require the use of external calibrator beads of known concentration that are spiked into the sample. However, this counting measurement can lead to an over-estimation of the sample concentration. Moreover, counting beads are quite expensive and increase the cost of the assay, particularly when a large number of samples should be processed. Another counting method is based on the use of the fluorescent DNA binding agent, 7-amino actinomycin D (7AAD), able to define dead and live populations by flow cytometry.

The purpose of this study was to compare the traditional manual counting method to the automatic counting assay based on the CytoFLEX instrument for their ability to measure white blood cells from peripheral blood (PB) of CML patients. In the automatic method, we investigated four different types of measurements: absolute cell-counting, a measurement based on beads, cell counting with the use of 7AAD day, and a combined method including counting beads and 7AAD. Our analyses focused on both qualitative and quantitative parameters, including total amount of WBCs, RNA recovery efficiency, and quantification of *BCR-ABL1* and *ABL1* gene transcript levels from matched samples of CML patients.

## 2. Materials and Methods

### 2.1. Patient Selection

The research study was carried out at the “Center of Experimental Oncology and Hematology” of the A.O.U. Policlinico “G. Rodolico-San Marco” of Catania. Between May 2021 and December 2021, peripheral blood samples from 90 adults’ chronic phase CML (CP-CML) patients were collected and analyzed for molecular monitoring of CML in our Molecular Diagnostic laboratory. Blood samples collection was conducted in accordance with the Declaration of Helsinki. All patients gave written informed consent prior to participation.

All patients received a TKI (TKI: imatinib brand or generic, dasatinib, nilotinib or bosutinib) as first-line treatment. Treatment response was assessed according to the 2020 ELN criteria [12].

### 2.2. Blood Collection and White Blood Cell Isolation

Ninety CML patients provided 28 mL of PB within a single blood draw, collected in sterile 4 × 7 mL ethylene-diamin tetra-acetic acid (EDTA) tubes (BD Vacutainer^®^, Becton Dickinson, Franklin Lanes, NJ, USA), according to the manufacturer’s instructions. After blood draw, blood samples were stored at room temperature and further processed within 24 h. Total WBC were isolated using the Biomek i-5 Automated Workstations integrated with the CytoFLEX Flow Cytometer (Beckman Coulter, Milano, Italia), as previously described [26]. Briefly, EDTA tubes with an Identificatory (ID) patient number were first logged in a datasheet and then loaded into the Biomeck i-5 system. Next, the whole blood was transferred to 50 mL tubes by an arm linked to a span-8 Pod. Red cells were removed by three consecutive red cell lysis treatments (10 min, 10 min and 5 min), followed by a centrifugation step (7 min @ 1800 rpm). Then, the entire white blood cells were collected in phosphate buffered saline (PBS) and a count was carried out.

### 2.3. White Blood Cells Count

Two count methods were investigated: the manual (A) and the automatic (B–E) methods. In the manual methods, samples were diluted and counted using a hemocytometer counting-chamber with a microscope. Next, 1 × 10^7^ cells were lysed in RLT buffer (Qiagen, Hilden, Germany), according to the manufacturers’ instructions (Figure 1). RLT lysates were stored at −80 °C until further processing.

In the automatic method, cells were loaded in a 96-well plate and cell-count was carried out in the CytoFLEX Flow Cytometer by four different types of measurements. In the first measurement (B), absolute cell-counting was performed using 200 µL of re-suspended cells in PBS buffer, as previously described [26]. The other three measurements were obtained using: (C) counting beads, (D) 7AAD solution, or (E) either counting beads and 7AAD (Figure 1). In the beads method, 100 µL counting beads were added, in a known concentration, to 100 µL of samples, mixed well and counted in the flow cytometer together with white blood cells. The absolute count of blood cells was then calculated as the product of the cell-to-bead count ratio and the concentration of counting beads. In the 7AAD measurement, a total of 10 µL fluorescent intercalator 7-AAD solution was added to 200 µL cells re-suspended in PBS and mixed. The cells were stained for 10 min at room temperature while protected from light and then loaded into the CytoFLEX Flow Cytometer. In the combined measurement, a solution with 100 µL of cells plus 100 µL of counting beads was made. Then, 10 µL of 7-AAD was added to the solution, mixed gently and incubated for 10 min at room temperature in the dark before loading into the flow cytometer. Overall, a total of 10,000 events were counted and cells present in 1 mL of PBS were calculated by reporting the results as “cell lives events/µL (V) × 1 mL”. Finally, 1 × 10^7^ cells were lysed in RLT buffer, as mentioned above. RLT lysates were stored at −80 °C until further processing.

### 2.4. RNA Extraction and cDNA Synthesis

Total RNA from matched samples was extracted from 1 × 10^7^ WBC lysed in RLT buffer by QIAsymphoy^®^ technology (Qiagen, Hilden, Germany), according to the manufacturers’ instructions, and RNA was eluted in Dnase/Rnase free water, as previously reported [29].

Purified total RNA was quantified with a BioSpectrometer (Eppendorf, Hamburg, Germany) at wavelengths of A230, A260 and A280 nanometer. RNA purity was calculated by A260/280 ratio (~1.9–2.0) and A260/230 ratio (~2.0–2.2). RNA integrity was then verified by electrophoresis running samples on 1.2% denaturing agarose gels. Total RNA samples were stored at −20 °C until further use.

Complementary DNA (cDNA) was synthetized from a total of 1 µg of purified RNA using random hexamer primers (Promega, Madison, WI, USA) and moloney murine leukemia virus reverse transcriptase (Thermo Fisher, Waltham, MA, USA) enzyme, as previously reported [30].

### 2.5. Quantification of BCR-ABL1 and ABL1 Transcripts

The *BCR-ABL1* and *ABL1* gene transcript levels from matched samples were quantified using real-time PCR (Q-PCR), at the Centre of Experimental Oncology and Hematology, as previously reported [31]. The *BCR-ABL1/ABL1* determination was assessed according to the international scale (IS), and calculated by ratio of *BCR-ABL1* and *ABL1* transcript levels. This value was expressed as percentage on a log scale and using a conversion factor (CF) calculated every year, as previously described [32]. The Q-PCR determinations were considered of appropriate quality only in the presence of no less than 10,000 *ABL1* copies [32,33,34]. Quantitative analysis of RNA isolated from matched samples was assessed comparing the cycle threshold (Ct) values achieved by Q-PCR for both *BCR-ABL1* and *ABL1* genes.

### 2.6. Software and Statistical Analyses

Cell count was performed using the CytoFLEX Flow Cytometer and the data analyzed employing CytExpert program (version 2.2.0.97—Beckman Coulter, Inc., Brea, CA, USA). The *t*-test was used to compare the difference between matched samples obtained with the five counting methods. Difference in number of white blood cells and Ct values (for *BCR-ABL1* and *ABL1* genes) were calculated and a *p* value below 0.05 was considered statistically significant. To evaluate the bias between the mean differences of the methods and to estimate an agreement interval within 95% interval, a Bland-Altman plot was used. For both *t*-test and Bland-Altman plot, Prism software v. 8.4 was employed.

## 3. Results

### 3.1. Patient Characteristics

Patient characteristics are summarized in Table 1. The median age of the accrued population was 63 years old (range 25–82) and median follow up was 60 months (range 5–105). Of total patients, 58.9% were male while 41.1% were female. The median leukocyte count was 9.85 × 10^9^/L (range 6.20–20.8) and the median of hemoglobin was 12.5 g/dL (range 10.8–14.5). Thirty-four patients showed an e13a2 (b2a2) *BCR-ABL1* fusion transcript, 48 subjects showed an e14a2 (b3a2) *BCR-ABL1* rearrangement and eight exhibited both e13a2 and e14a2 isoforms. According to *BCR-ABL1^IS^* transcript levels, we selected patients with a molecular response distributed in three groups of 30 subjects each: Group A (10% > *BCR-ABL1/ABL1^IS^* > 1%), Group B (1% ≥ *BCR-ABL1/ABL1^IS^* > 0.1%) and Group C (0.1% ≥ *BCR-ABL1/ABL1^IS^* > 0.01%).

### 3.2. Comparison of Count Efficiency and RNA Isolation by Five Different Measurement Methods

In order to compare the count methods efficiency of the five measurements, we evaluated the count of WBCs using Manual, Automatic, Automatic + Beads, Automatic + 7AAD and Automatic + Beads + 7AAD methods (Figure 2). We observed comparable count efficiency between the methods and, interestingly, no statistical differences were observed between the five counting assays (Figure 2). The median of counted cells was 2.46 × 10^6^ (range 1.20 × 10^6^–6.30 × 10^6^) with the Manual protocol, 2.46 × 10^6^ (range 1.20 × 10^6^–6.27 × 10^6^) with the Automatic method, 2.60 × 10^6^ (range 1.26 × 10^6^–6.06 × 10^6^) when we measured the cells with Automatic and Beads measurement, 2.43 × 10^6^ (range 1.22 × 10^6^–6.16 × 10^6^) when we used the fluorescent intercalator 7-AAD solution and, finally, 2.47 × 10^6^ (range 1.20 × 10^6^–6.04 × 10^6^) when we combined Automatic count plus Beads and 7AAD solution (Table 2, cell isolation).

Next, to evaluate the impact on the downstream analysis for molecular monitoring of CML patients, we compared the quantity and quality of RNA extracted from matched RLT samples provided by five different measurements. We observed that the median of RNA concentration was similar in all used methods (Table 2, RNA isolation). In detail, RNA concentration expressed as ng/μL was 123.00 (range 80–223.5) when we used the Manual method, 120.35 (range 75–256.4) in samples counted by automatic measurement, 118.12 (range 75–200.4) using the beads method, 126.00 (range 78–232.5) in the 7AAD assay and 116.65 (range 75–223.3) combining the beads with 7AAD solution. Overall, we obtained a good RNA quality from matched samples as measured by RNA spectrophotometric quantification at wavelengths of A_260/280_ (1.9 median value) and A_260/230_ (2.1 median value).

### 3.3. Concordance of Cycle Threshold Values for BCR-ABL1 and ABL1 Genes According to the Five Counting Methods

Previous evidence has established that the measurement of *BCR-ABL1/ABL1^IS^* transcript levels may be affected by the accuracy of the method used [21,24]. Variables of different procedures may result in different molecular response scoring for CML patients. The choose of counting method is crucial to obtain high-quality and reproducible data. To this purpose, we stratified the CML patients according to their *BCR-ABL1/ABL1^IS^* transcript level into three groups (Group A, B and C) and then we compared Ct value obtained from matched specimens counted by the five different methods. By considering the *BCR-ABL1* Q-PCR data, we found similar Ct values in all investigated groups. Specifically, we observed that in Group A the Ct value median was 29.30 (range 25.52–31.01) for samples counted by hemocytometer counting-chamber, 29.28 (range 25.66–31.10) for the automatic method, 29.37 (range 25.89–31.55) using counting beads, 29.24 (range 25.50–31.08) adding only the intercalator 7AAD and 29.34 (range 25.56–31.06) combining the counting beads with 7AAD solution (Figure 3A). When we looked at Group B, we detected a Ct value median of 32.01 (range 29.85–37.85) in the manual method, 32.03 (range 29.53–37.99) in automatic measurement, 32.10 (range 30.12–37.22) in presence of counting beads, 32.04 (range 29.89–37.10) using 7AAD solution and 32.05 (range 30.02–37.15) in the combined assay (Figure 3B). In the last Group (C), we observed a Ct value median of 36.30 (range 33.16–38.55) by manual count, 36.31 (range 33.18–38.42) in the automatic method, 36.36 (range 33.26–38.42) with counting beads, 36.30 (range 33.17–38.82) in presence of 7AAD solution and 36.36 (range 33.28–38.44) using counting beads plus the intercalator 7AAD (Figure 3C).

The Bland-Altman plot showed no consistent bias between the manual and the automatic methods (Figure 4A–C). Moreover, no consistent bias was found when we repeated the analysis comparing the manual and the other three measurements (manual vs counting beads, manual vs intercalator 7AAD methods and manual vs the combination of beads with 7AAD solution measurements) (Appendix A).

Next, we investigated the *ABL1* Q-PCR data and did not observe statistically significant differences in the *ABL1* gene Ct values of matched samples evaluated for the five counting methods (Figure 5A–C). Furthermore, evaluating the *ABL1* reference gene copies we found that all measurement assay performed optimally with *ABL1* gene copies measured >10.000 in the three groups (data not showed). Overall, we found a 100% concordance. 

In this case also, the Bland-Altman plot showed no consistent bias between samples counted by the manual vs the other four methods (Figure 6A–C and Appendix A).

## 4. Discussion and Conclusions

The molecular monitoring of CML patients has become of pivotal importance to predict treatment response and relapse [22,27]. Quantitative PCR analyses of blood samples, assessing the level of the *BCR-ABL1* oncogene relative to an internal control gene (usually *ABL1*), are the “gold standard” for monitoring the kinetics of disease burden variation. Moreover, molecular detection of *BCR-ABL1* transcript by Q-PCR has the advantages of reliability, sensitivity and reproducibility of results.

Upon TKI therapy initiation, *BCR-ABL1* molecular testing is generally recommended every 3 months as far as a stable deep molecular response is achieved. Currently, more frequent molecular testing is encouraged for CML individuals achieving persistent deep molecular response after TKI discontinuation (a condition defined as TFR) [18,35]. Moreover, a more strictly molecular monitoring ensuring a timely recognition of rising *BCR-ABL1* values should trigger a thorough evaluation regarding compliance or possible relapse [36]. To achieve this effectively, standardization of the laboratory procedures is mandatory, including method for cell counting and isolation. 

Nowadays, many cell counting assays enable the detection and enumeration of cells, either manually or via an automated process. In order to obtain reliable results, the entire cell counting process should be standardized, as variability of the measurements may introduce unintended error. Despite the easy procedure, the manual method is laborious can process a small number of samples at a time and is less suitable for large-scale studies or in diagnostic practice. Here, we present a comparison between a traditional counting method and our automated system based on the CytoFLEX Flow Cytometer integrated with the semiautomatic Biomek i-5 Workstation. Overall, the automatic counting system determined a hands-on time reduction allowing processing of a large number of samples concurrently (up to 20 specimens in only five minutes). Samples are loaded in a 96-well plate and then a peristaltic pump system permits recording of an unlimited number of patients, without any time delay or dilution influence. Furthermore, the opportunity to manipulate the live cell gate on the CytoFLEX software enabled a more reliable count.

Different studies have recommended the addition of counting beads, at a known concentration, to the cells before measurement in order to calculate cell densities through flow cytometry [37,38,39]. However, as beads tend to stick to the plastic tube, this method might overestimate the sample concentration. For this purpose, in our study we compared the automatic count in the presence or not of beads (absolute count). Among the tested methods, we did not observe consistent bias. Of note, the absolute count allows a decrease in the assay cost per run compareded to the expensive beads cost, without affecting the quality and quantity of RNA for downstream analyses. 

Staining with the intercalator 7AAD dye has the advantage of discriminating dead from live cell population, even with the limitation of more time for staining. Therefore, we also compared the absolute count to the measurement based on 7AAD solution. Again, no differences were observed among the counting assays, suggesting that the 7AAD intercalator does not provide additional information, either used alone or in combination with counting beads.

Several reports have shown that the quality and quantity of the *BCR-ABL1/ABL1* transcript level may influence the molecular outcome for CML patients [9,24]. Particularly, it is advised that a sample should have at least 10,000.00 *ABL1* gene copies to pass minimum quality standards. Furthermore, the amount of the control gene is critical to define the kinetics of the disease burden reduction. In this context, we also compared the Ct values from matched counted samples obtained by Q-PCR assay on *BCR-ABL1* and *ABL1* genes and observed similar quantitative results as well as recovery efficiency. Likewise, matched samples were correctly classified into the same group of molecular response. These data underline the importance of taking into consideration the counting method used (manual or automatic) and readout (*BCR-ABL1* and *ABL1* transcript levels) when comparing results for diagnostic practice.

In conclusion, the automatic system based on absolute cell counting allows the obtaining of accurate cell counts without the use of expensive beads or the addition of the time-consuming intercalator 7AAD, and can streamline the entire procedure. Therefore, this measurement can replace the more laborious manual workflow, largely based on an interaction with the users, especially when high-throughput isolations from the blood of CML patients are needed.

## Figures and Tables

**Figure 1 diagnostics-12-01051-f001:**
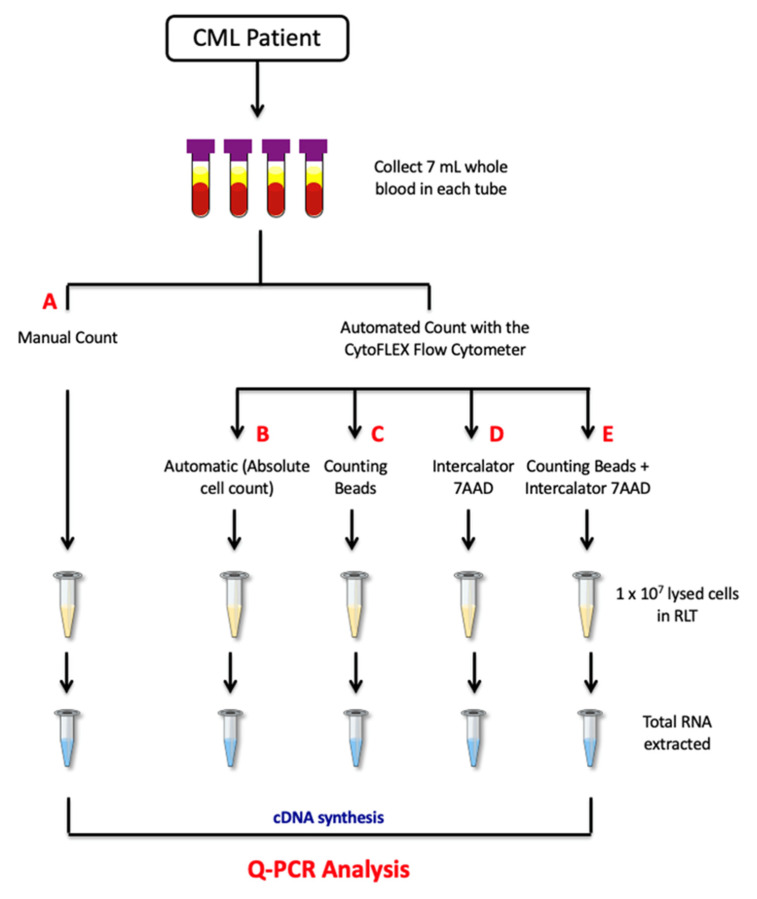
Workflow of the study. A total of 28 mL of peripheral blood from CML patients was collected within a single blood draw, in sterile 4 × 7 mL EDTA tubes. Total white blood cells (WBC) were isolated using the Biomek i-5 Automated Workstations and re-suspended in phosphate buffered saline (PBS). Next, a count was carried out by the manual (**A**) and the automatic methods via the CytoFLEX instrument (**B**–**E**). In the manual methods, samples were diluted and counted using the hemocytometer counting-chamber with a microscope. In the automatic method, four different measurements were tested: the automatic absolute protocol, the automatic assay using counting beads, the automatic protocol with 7-Aminoactinomycin Dye (7AAD) solution, and the automatic method based on the use of either counting beads and 7AAD. Next, 1 × 10^7^ of the collected cells were lysed in RLT buffer and total RNA was isolated from matched samples. Finally, quantitative polymerase chain reaction (Q-PCR) was used to measure *BCR-ABL1* and *ABL1* gene transcript levels. EDTA tube: EthylenDiaminoTetracetyc Acid tube; RNA: RiboNucleic Acid; Q-PCR: quantitative polymerase chain reaction; cDNA: complementary DeossiNucleic Acid.

**Figure 2 diagnostics-12-01051-f002:**
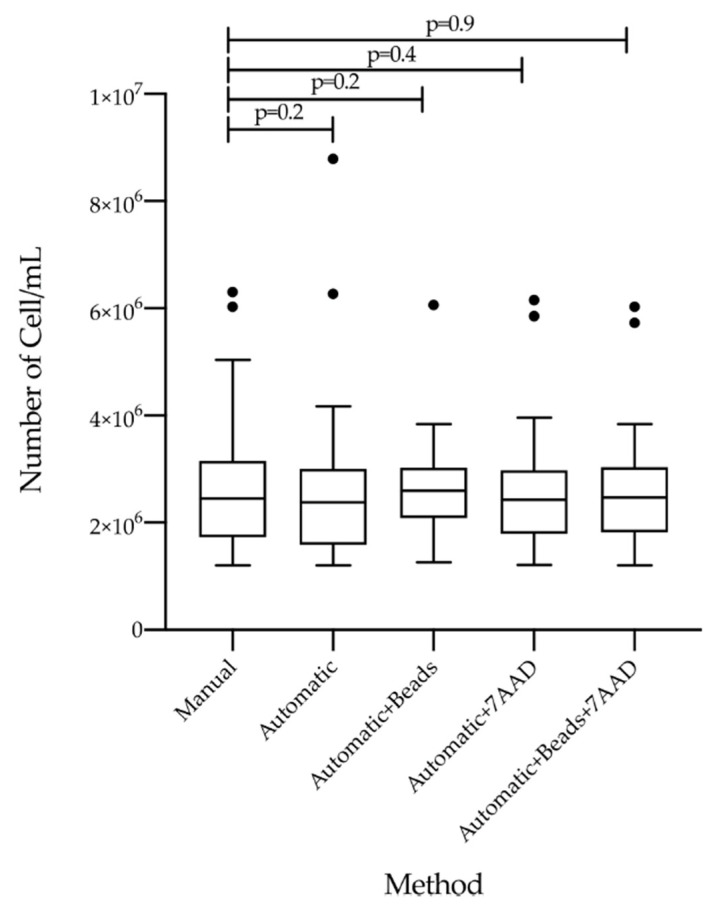
Comparison of five counting methods respect to the number of white blood cells. White blood cells were isolated from 28 mL peripheral blood samples of 90 patients with chronic myeloid leukemia (CML). Cells were enumerated using five different counting methods: the manual method, the automatic absolute measurement, the automatic assay using counting beads, the automatic protocol with 7-Aminoactinomycin D (7AAD) solution, and the automatic method based on the use of either counting beads and 7AAD. Cell counts, expressed as number of cells/mL solution, were investigated to determine the counting recovery efficiency. The number of cells was determined for each method and showed as Tukey-boxplots. Thick lines in each boxplot represent the median of number of cells/mL for each method. The dark dots indicate the outlier’s value. The student-paired *t*-test was used to test the difference between the five counting methods and *p* value below 0.05 was considered statistically significant.

**Figure 3 diagnostics-12-01051-f003:**
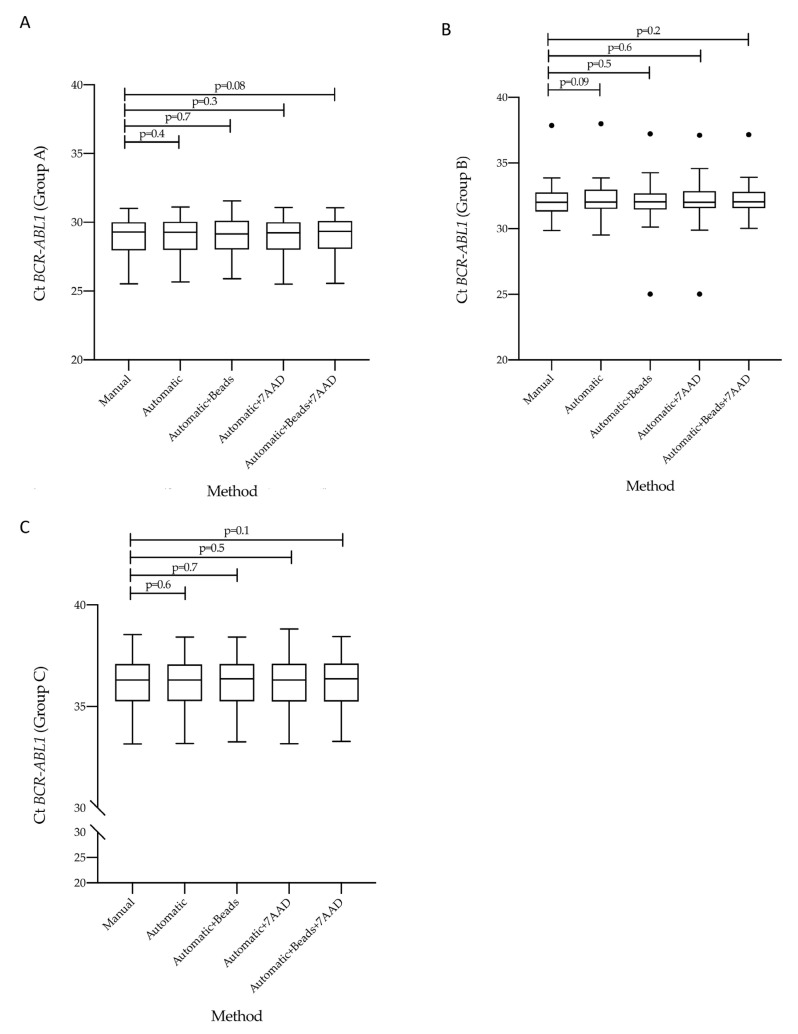
Measurement of *BCR-ABL1* Cycle threshold value on matched samples counted by five different enumeration assays. Comparison of the Cycle Threshold (Ct) values of *BCR-ABL1* gene transcript measured by Q-PCR in matched samples counted by five different protocols. *BCR-ABL1* gene transcript was assessed in patients stratified into three groups, each consisting of 30 individuals, according to their *BCR-ABL1/ABL1^IS^* transcript: Group A (10% > *BCR-ABL1/AB1L^IS^* > 1%) (**A**), Group B (1% > *BCR-ABL1/ABL1^IS^* > 0.1%) (**B**), and Group C (0.1% > *BCR-AB1L/ABL1^IS^* > 0.01%) (**C**). The *BCR-ABL1* Ct values were determined for each method and showed as Tukey-boxplots. Thick lines in each boxplot represent the median *BCR-ABL1* Ct value for each counting method. The dark dots indicate the outlier’s values. The student-paired *t*-test was used to test the difference between the five counting methods and *p* value below 0.05 was considered statistically significant.

**Figure 4 diagnostics-12-01051-f004:**
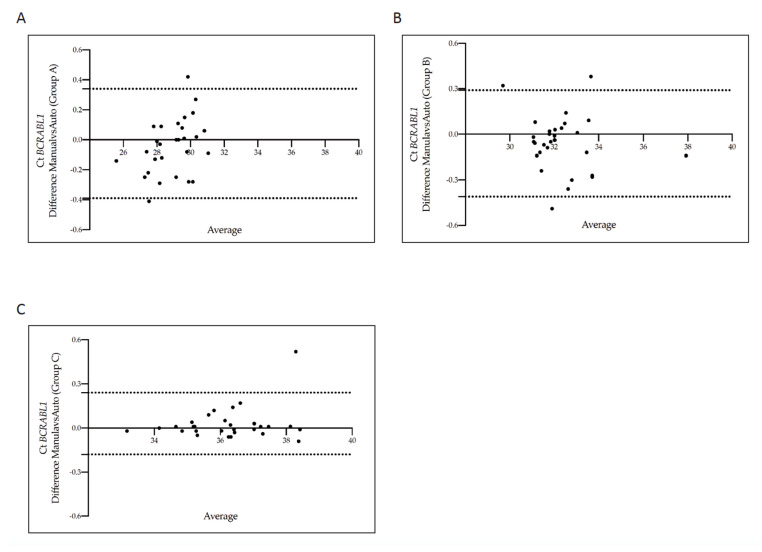
Bland–Altman showing the concordance of the *BCR-ABL1* Cycle threshold values measured in matched samples counted by the manual and the automatic absolute cell-counting assay. Paired measurements of *BCR-ABL1* Ct value were combined for patients stratified into three groups, each consisting of 30 individuals: Group A (10% > *BCR-ABL1/ABL^IS^* > 1%) (**A**), Group B (1% > *BCR-ABL1/ABL1^IS^* > 0.1%) (**B**), and Group C (0.1% > *BCR-ABL1/ABL^IS^* > 0.01%) (**C**). The graph is plotted on the XY axis where X depicts the difference of the two measurements, and the Y-axis shows the mean of the two measurements. Horizontal lines are drawn at the mean difference between the two counting methods and the upper and lower limits of agreement. The 95% confidence intervals are shown for the mean and the upper and lower limits of agreement.

**Figure 5 diagnostics-12-01051-f005:**
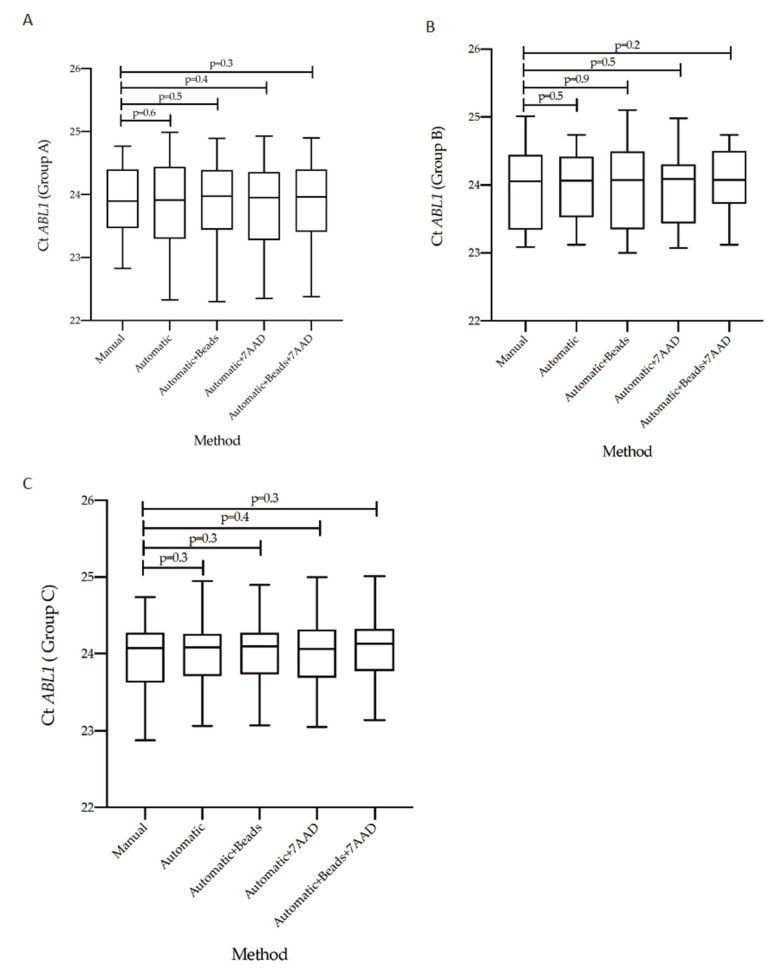
*Measurement of Cycle threshold ABL1 value on matched samples counted by five different enumeration assays.* Comparison of the Cycle Threshold (Ct) values of *ABL1* gene transcript measured by Q-PCR in matched samples counted by five different protocols. *ABL1* gene transcript was assessed in patients stratified into three groups, each consisting of 30 individuals, according to their *BCR-ABL1/ABL1^IS^* transcript: Group A (10% > *BCR-ABL1/ABL1^IS^* > 1%) (**A**), Group B (1% > *BCR-ABL1/ABL1^IS^* > 0.1%) (**B**), and Group C (0.1% > *BCR-ABL1/ABL1^IS^* > 0.01%) (**C**). The *ABL1* Ct values were determined for each method and showed as boxplots delimited by the 25th (lower) and 75th (upper) percentile. Horizontal lines above and below each boxplot indicate the 5th and 95th percentile, respectively. Thick lines in each boxplot represent the median *ABL1* Ct value for each counting method. The student-paired *t*-test was used to test the difference between the five counting methods and *p* value below 0.05 was considered statistically significant.

**Figure 6 diagnostics-12-01051-f006:**
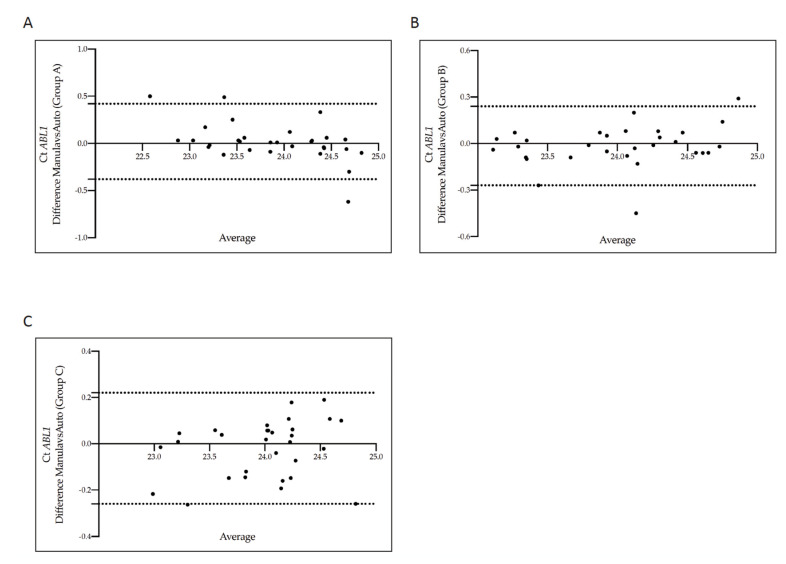
Bland–Altman showing the concordance of the *ABL1* Cycle threshold values measured in matched samples counted by the manual and the automatic absolute cell-counting assays. Paired measurements of *ABL1* Ct value were combined for patients stratified into three groups, each consisting of 30 individuals: Group A (10% > *BCR-ABL1/ABL1^IS^* > 1%) (**A**), Group B (1% > *BCR-ABL1/ABL1^IS^* > 0.1%) (**B**), and Group C (0.1% > *BCR-ABL1/ABL1^IS^* > 0.01%) (**C**). The graph is plotted on the XY axis where X depicts the difference of the two measurements, and the Y-axis shows the mean of the two measurements. Horizontal lines are drawn at the mean difference between the two counting methods and the upper and lower limits of agreement. The 95% confidence intervals are shown for the mean and the upper and lower limits of agreement.

**Table 1 diagnostics-12-01051-t001:** Patient Characteristics (*N* = 90).

Characteristics	N.
**Age**	
Median	63
Range	25–82
**Follow up**	
Median (mo.)	60
Range	5–105
**Sex (pts n.)**	
Male	53 (58.9%)
Female	37 (41.1%)
**Leukocyte count (×10^9^/L)**	
Median	9.85
Range	6.20–20.8
**Platelet count (×10^9^/L)**	
Median	350
Range	80–758
**Hemoglobin (g/dL)**	
Median	12.5
Range	10.8–14.5
**Transcript Type**	
e13a2 (b2a2)	34
e14a2 (b3a2)	48
e13a2 and e14a2	8
**Molecular response**	
**GROUP A** (10% > *BCR-ABL1/ABL1^IS^* > 1%)	30
**GROUP B** (1% ≥ *BCR-ABL1/ABL1^IS^* > 0.1%)	30
**GROUP C** (0.1% ≥ *BCR-ABL1/ABL1^IS^* > 0.01%)	30

**Table 2 diagnostics-12-01051-t002:** Comparison of the amount of white blood cells and RNA isolated in matched samples by five different counting methods.

	Cells Isolation		RNA Isolation		
Protocol	Cells/mL Median Range	Total Cells Median Range	ng/µL Median Range	260/280 Median Range	260/230 Median Range
Manual	2.46 × 10^6^ (1.20 × 10^6^–6.30 × 10^6^)	1.23 × 10^8^ (6.00 × 10^7^–3.15 × 10^8^)	123.00 (80–223.5)	1.9 (1.9–2.0)	2.1 (2.0–2.2)
Automatic (PBS with cells)	2.46 × 10^6^ (1.20 × 10^6^–6.27 × 10^6^)	1.23 × 10^8^ (6.00 × 10^7^–3.14 × 10^8^)	120.35 (75–256.4)	1.9 (1.90–2.0)	2.1 (2.0–2.2)
Automatic + beads	2.60 × 10^6^ (1.26 × 10^6^–6.06 × 10^6^)	1.30 × 10^8^ (6.30 × 10^7^–3.03 × 10^8^)	118.12 (75–200.4)	1.9 (1.90–2.0)	2.1 (2.0–2.2)
Automatic + 7AAD	2,43 × 10^6^ (1.22 × 10^6^–6.16 × 10^6^)	1.22 × 10^8^ (6.10 × 10^7^–3.08 × 10^8^)	126.00 (78–232.5)	1.9 (1.9–2.0)	2.1 (2.0–2.2)
Automatic + 7AAD + beads	2,47 × 10^6^ (1.20 × 10^6^–6.04 × 10^6^)	1.24 × 10^8^ (6.00 × 10^7^–3.02 × 10^8^)	116.65 (75–223.3)	1.9 (1.90–2.0)	2.1 (2.0–2.2)

## Data Availability

Data is contained within the article or Appendix A.

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
