# Peer review of "Impact of Different Cell Counting Methods in Molecular Monitoring of Chronic Myeloid Leukemia Patients"

_diagnostics, 2022, doi:10.3390/diagnostics12051051_

Round 1

Reviewer 1 Report

The authors reported a study of comparison of five different counting methods in 90 patients with chronic myeloid leukemia to monitor BCR/ABL1 and ABL1 transcripts. The study is important for the purpose of optimizing  cell counting methods by allowing faster screening. Among four flow cytometry-based counting assays, the authors found comparable results by using the flow cytometry-based absolute counting. 

The manuscript is very clear and the research has been well conducted. The methods are well described and although of routine, they provide a useful diagnostic view.

My comments are below:

  1. Were the outliner values the same in different counting methods? Was the counting of cells of the same patient always an outliner in different methods?
  2. Were the authors able to define a sensitivity cut off by which one method will be better than another?

Author Response

Author’s Replay to the Reviewer 1:

  1. Were the outliner values the same in different counting methods? Was the counting of cells of the same patient always an outliner in different methods?

Authors Response:

We checked the outliner values for Figures 3 and 5 and we found that they were relative to the same patient measured with the different counting methods.

  1. Were the authors able to define a sensitivity cut off by which one method will be better than another?

Authors Response:

The different counting methods showed similar median Ct values associated to number of counted white blood cells, indicating that the sensitivity cut off is similar among the five tested methods. On the basis of this observation, we demonstrated that FC-based automatic absolute cell counting is suitable due to its less expensive and easy protocol.

Reviewer 2 Report

In this paper, the authors compared methods for cell counting for patients with CML. All methods have performed similarly to manual calculation. Their conclusion was that the flow cytometry-based method is cheaper and less laborious than other methods.

Grammar could benefit from a second look.

Author Response

Author’s Replay to the Reviewer 2:

  1. Grammar could benefit from a second look.

Authors Response:

We agree with the reviewer’s observation and we performed an extensive revision of our manuscrpt and corrected the grammar and mispelling errors.
